# ADCC against MICA/B Is Mediated against Differentiated Oral and Pancreatic and Not Stem-Like/Poorly Differentiated Tumors by the NK Cells; Loss in Cancer Patients due to Down-Modulation of CD16 Receptor

**DOI:** 10.3390/cancers13020239

**Published:** 2021-01-11

**Authors:** Kawaljit Kaur, Tahmineh Safaie, Meng-Wei Ko, Yuhao Wang, Anahid Jewett

**Affiliations:** 1Division of Oral Biology and Oral Medicine, School of Dentistry and Medicine, Los Angeles, CA 90095, USA; drkawalmann@g.ucla.edu (K.K.); tahmineh19521@g.ucla.edu (T.S.); mengwei@g.ucla.edu (M.-W.K.); yuhaowang@ucla.edu (Y.W.); 2The Jonsson Comprehensive Cancer Center, UCLA School of Dentistry and Medicine, Los Angeles, CA 90095, USA

**Keywords:** NK cells, cancer stem cells (CSCs), antibody-dependent cellular cytotoxicity (ADCC), differentiation, cytotoxicity, IFN-γ, osteoclasts, MICA/B mAb

## Abstract

**Simple Summary:**

Natural Killer cells are known to eliminate tumors directly or via antibody dependent cellular cytotoxicity. The complete modes and mechanisms of such killings are yet to be delineated. It is also unclear at what stages of tumor differentiation NK cells are capable of mediating the two modes of tumor killing. In this report we provide evidence that NK cells mediate killing of both stem-like/poorly differentiated tumors and well-differentiated tumors via direct cytotoxicity and antibody dependent cellular cytotoxicity, respectively. By using antibodies to MICA/B, EGFR and PDL1 surface receptors expressed on well-differentiated but not on stem-like/poorly differentiated tumors we demonstrate significant NK cell mediated antibody dependent cellular cytotoxicity in the absence of direct killing. In addition, our results suggested the possibility of CD16 receptors mediating both direct cytotoxicity and antibody dependent cellular cytotoxicity, resulting in the competitive use of these receptors in either direct killing or antibody dependent cellular cytotoxicity.

**Abstract:**

Tumor cells are known to upregulate major histocompatibility complex-class I chain related proteins A and B (MICA/B) expression under stress conditions or due to radiation exposure. However, it is not clear whether there are specific stages of cellular maturation in which these ligands are upregulated or whether the natural killer (NK) cells differentially target these tumors in direct cytotoxicity or antibody-dependent cell cytotoxicity (ADCC). We used freshly isolated primary and osteoclast (OCs)-expanded NK cells to determine the degree of direct cytotoxicity or of ADCC using anti-MICA/B monoclonal antibodies (mAbs) against oral stem-like/poorly-differentiated oral squamous cancer stem cells (OSCSCs) and Mia PaCa-2 (MP2) pancreatic tumors as well as their well-differentiated counterparts: namely, oral squamous carcinoma cells (OSCCs) and pancreatic PL12 tumors. By using phenotypic and functional analysis, we demonstrated that OSCSCs and MP2 tumors were primary targets of direct cytotoxicity by freshly isolated NK cells and not by ADCC mediated by anti-MICA/B mAbs, which was likely due to the lower surface expression of MICA/B. However, the inverse was seen when their MICA/B-expressing differentiated counterparts, OSCCs and PL12 tumors, were used in direct cytotoxicity and ADCC, in which there was lower direct cytotoxicity but higher ADCC mediated by the NK cells. Differentiation of the OSCSCs and MP2 tumors by NK cell-supernatants abolished the direct killing of these tumors by the NK cells while enhancing NK cell-mediated ADCC due to the increased expression of MICA/B on the surface of these tumors. We further report that both direct killing and ADCC against MICA/B expressing tumors were significantly diminished by cancer patients’ NK cells. Surprisingly, OC-expanded NK cells, unlike primary interleukin-2 (IL-2) activated NK cells, were found to kill OSCCs and PL12 tumors, and under these conditions, we did not observe significant ADCC using anti-MICA/B mAbs, even though the tumors expressed a higher surface expression of MICA/B. In addition, differentiated tumor cells also expressed higher levels of surface epidermal growth factor receptor (EGFR) and programmed death-ligand 1(PDL1) and were more susceptible to NK cell-mediated ADCC in the presence of anti-EGFR and anti-PDL1 mAbs compared to their stem-like/poorly differentiated counterparts. Overall, these results suggested the possibility of CD16 receptors mediating both direct cytotoxicity and ADCC, resulting in the competitive use of these receptors in either direct killing or ADCC, depending on the differentiation status of tumor cells and the stage of maturation and activation of NK cells.

## 1. Introduction

Natural killer (NK) cells were first discovered as a functional cell type in 1970 and were named by Kiessling et al. in 1975 [1]. NK cells were so named for their effector functions, which include direct natural cytotoxicity, antibody-dependent cellular cytotoxicity (ADCC), as well as the secretion of inflammatory cytokines and chemokines which indirectly regulate the functions of other immune cells [2,3]. Conventional human NK cells are identified by the expression of CD16 and CD56 and by the lack of surface CD3 receptor expression [4]. NK cells mediate their functions through several important activating and inhibitory cell receptors such as CD16, NKG2D, natural cytotoxicity receptors (NCR), killer immunoglobulin-like receptors (KIR), and the NKG2 family of receptors, which form heterodimers with CD94 [5,6,7]. The balance between activating and inhibitory signals that NK cells receive through the surface receptors determines their functional fate [5]. As such, activated NK cells are able to recognize and lyse tumor cells expressing certain surface receptors without prior antigenic sensitization [8,9]. Many tumors, especially differentiated tumors, express major histocompatibility complex-class I (MHC-class I) chain related proteins A and B (MICA/B), which mark them for elimination by the NK cells [10,11,12,13]. However, tumor cells can successfully evade detection by NK cells by shedding MICA/B [11,14]. Moreover, differentiated tumors were also found to have higher expression of epidermal growth factor receptor (EGFR) [15,16] and programmed death-ligand 1(PDL1) [17,18].

Studies have shown that NK cell-mediated ADCC can be exploited as an important cancer treatment [19]. NK cell-mediated ADCC is triggered when FcγRIIIA (CD16) binds to the Fc region of antibodies bound to their cognate antigens expressed on target cells. This binding induces the directed exocytosis of granzyme- and perforin-containing granules that then lyse the target cells [20,21]. Thus, CD16 is a major FcγR on NK cells and is crucial for activating ADCC activity in NK cells [22,23,24,25]. NKG2D is an activating surface receptor of NK cells which in conjunction with CD16 influences NK cell function [26,27,28]. In addition, NKG2D was found to play a significant role in tumor rejection and tumor immunosurveillance through binding to MICA/B, which are among the ligands binding to NKG2D receptors [29,30,31,32]. However, tumor-associated NK cells are refractory to CD16 receptor stimulation, resulting in diminished ADCC against autologous tumor cells [33]. Moreover, ADCC was also found to be impaired in cancer patients [34,35,36,37].

We have previously demonstrated that NK cells secrete elevated levels of cytokines, particularly interferon gamma (IFN-γ) and tumor necrosis factor alpha (TNF-α), in the presence of decreased cytotoxicity when CD16 receptors are triggered on their surface. We termed this functional stage of NK cells as “split anergy”. As indicated, NK cells become split-anergized upon CD16 receptor crosslinking or during interactions with cancer stem cells (CSCs) or undifferentiated cells [38,39]. Cytokines secreted by split-anergized NK cells play an important role in mediating tumor cell differentiation [38,40,41]. It was previously demonstrated that decreased NK cell counts, suppressed NK function, and the down-modulation of NK cell surface receptors in the peripheral blood and the tumor microenvironment were associated with poor prognoses in cancer patients [42,43,44,45,46,47,48,49,50,51,52,53,54,55,56,57,58,59,60]. Our previous work also illustrated that NK cells of cancer patients and of tumor-bearing humanized-bone marrow/liver/thymus (BLT) mice mediated less cytotoxicity against cancer cells and secreted lower levels of cytokines [61,62,63].

This study explored the different levels of NK cell-mediated IFN-γ secretion, direct cytotoxicity, ADCC, and the surface receptor expression on NK cells from healthy individuals and those of cancer patients. Next, we determined the surface expression of MICA/B on CSCs and their differentiated counterparts. Finally, we elucidated the differences between the NK cell-mediated ADCC between freshly isolated NK cells, osteoclast (OC)-expanded supercharged NK cells, and NK92 tumors transfected with CD16 receptors. NK cell-mediated ADCC against MICA/B bearing differentiated tumor cells in the presence of MICA/B antibody was compared to that mediated by the antibodies against EGFR and PDL1.

## 2. Results

### 2.1. Cancer Patients’ NK Cells Exhibit Decreased Direct Killing and NK Cell-Mediated ADCC Compared to Healthy Individuals’ NK Cells

We determined NK cell function in cancer patients using NK cell-mediated cytotoxicity and IFN-γ secretion. We found that cancer patients’ NK cells mediated significantly lower levels of cytotoxicity (Figure 1A and Appendix A) and secreted lower amounts of IFN-γ when compared to healthy individuals’ NK cells (Figure 1B and Appendix A). Decreased IFN-γ secretion by patients’ NK cells was also seen in the presence of CSCs (Appendix A). Then, we used interleukin-2 (IL-2)-treated NK cells from cancer patients and healthy individuals as effectors to target tumors in the absence and presence of anti-MICA/B monoclonal antibodies (mAbs). We observed NK cells mediated ADCC against differentiated tumors (Figure 1D,E,G,H) but not against stem-like tumors (Figure 1C,F). However, very little or no NK cell-mediated ADCC against both oral (Figure 1C–E) and pancreatic tumor cells (Figure 1F–H) was seen from cancer patients’ NK cells. Next, we analyzed the expression of NK cell surface receptors isolated from healthy individuals and cancer patients. We detected a lower expression of Nkp44, CD94, NKG2D, and KIR2, and a higher expression of Nkp30, Nkp46, and KIR3 on CD16+ NK cells from cancer patients (Figure 1I–K and Appendix A). These data indicate that cancer patients’ NK cells express lower levels of surface receptors important in ADCC and substantially decreased cytotoxic activity against tumors when compared to those from healthy individuals’ NK cells.

### 2.2. Differentiated Tumor Cells Expressed Higher Levels of Surface MICA/B and Were More Susceptible to NK Cell-Mediated ADCC in the Presence of Anti-MICA/B mAb Compared to Their Stem-Like/Poorly Differentiated Couterparts

We have previously demonstrated that IFN-γ secreted by IL-2+anti-CD16 mAb-treated NK cells promotes tumor differentiation [18,64]. Therefore, we used the supernatants from IL-2+anti-CD16 mAb-treated NK cells to differentiate CSCs as described in the Materials and Methods section. We first investigated the surface expression of MICA/B on stem-like oral stem-like/poorly-differentiated oral squamous cancer stem cells (OSCSCs) and MP2 tumors, differentiated oral squamous carcinoma cells (OSCCs) and PL12 tumors, and NK cell-differentiated OSCSCs and MP2 tumors. OSCCs and the NK cell-differentiated OSCSCs expressed higher levels of MICA/B when compared to their stem-like counterparts OSCSCs (Figure 2A). Similarly, the differentiated pancreatic PL12 tumors and the NK cell-differentiated MP2 tumors expressed higher levels of MICA/B when compared to their stem-like counterparts MP2 tumors (Figure 2B).

Our previous studies also demonstrated that CSCs/poorly differentiated tumors are excellent targets of NK cell-mediated cytotoxicity, whereas their differentiated counterparts are significantly more resistant [64,65,66]. Here, we evaluated NK cell-mediated cytotoxicity against CSCs/poorly differentiated and well-differentiated tumor cells treated with monoclonal antibodies specific for MICA/B. We found that susceptibility to NK cell-mediated cytotoxicity increased significantly against anti-MICA/B mAb-treated OSCCs (Figure 2C,E,F), while anti-MICA/B mAb-treated OSCSCs remained relatively unchanged when compared to the killing of target cells in the absence of MICA/B antibodies (Figure 2D). Significantly higher levels of fold increase in NK cell-mediated ADCC were seen against OSCCs using MICA/B antibodies as compared to those in the absence of antibody, whereas slight increases in ADCC could be seen against OSCSCs when untreated or IL-2 treated NK cells were used as effectors (Figure 2H,I). The addition of antibody to CD16 receptor abolished the increase in ADCC against OSCCs (Figure 2G,J). Consistent with our findings in oral tumor cells, we observed higher levels of NK cell-mediated ADCC against anti-MICA/B mAb-treated pancreatic PL12 cells when compared to MP2 tumors (Figure 2K–O).

### 2.3. Differentiated Tumor Cells Treated with Anti-MICA/B mAb Triggered Increased IFN-γ Secretion by NK Cells

We have previously demonstrated that NK cells secrete higher levels of IFN-γ when co-cultured with CSCs/poorly differentiated tumor cells than with the differentiated tumors [64]. To assess the effect of anti-MICA/B mAbs-induced NK cell activation, we treated OSCSCs and OSCCs with or without anti-MICA/B mAbs and co-cultured them with NK cells. As expected, we did not detect any changes in IFN-γ secretion in cultures containing untreated NK cells (Figure 3A). When NK cells were treated with IL-2, we observed an increase in IFN-γ secretion in response to anti-MICA/B mAb-treated OSCCs as compared to untreated OSCCs (Figure 3B,D). There was no or a slight increase in IFN-γ secretion when IL-2-treated NK cells were co-cultured with anti-MICA/B mAb-treated OSCSCs as compared to untreated OSCSCs (Figure 3B,D). Consistent with our previous findings, the levels of IFN-γ produced by IL-2+anti-CD16 mAb treated NK cells were plateaued (Figure 3C). Therefore, we were not able to detect a noticeable difference in the concentrations of IFN-γ when untreated or anti-MICA/B mAb-treated OSCCs or OSCSCs were co-cultured with IL-2+anti-CD16 mAb-treated NK cells (Figure 3C).

### 2.4. Higher Levels of NK Cell-Mediated ADCC Were Seen in Freshly Isolated Primary NK Cells When Compared to OC-Expanded NK Cells

We have previously shown that osteoclast (OC)-expanded NK cells exhibit a greater potential to directly kill tumor cells while also expressing lower levels of CD16 receptors when compared to freshly isolated primary NK cells [61]. Thus, we tested surface expressions and NK cell-mediated ADCC from both freshly isolated primary NK cells and those expanded by the OCs. Primary NK cells expressed higher levels of CD16 but low no or low levels of NKG2D on their surface compared to OC-expanded NK cells (Appendix A). High levels of NK cell-mediated ADCC were seen in IL-2-treated primary NK cells, but very little or no NK cell-mediated ADCC was seen in OC-expanded NK cells (Figure 4A,B and Appendix A). Next, we determined the effects of CD16 cross-linking on primary and OC-expanded NK cells. As expected, in primary NK cells, NK cell-mediated cytotoxicity decreased when they were treated with IL-2+anti-CD16 mAb as compared to IL-2-activated NK cells (Figure 4C and Appendix A). However, we observed no significant differences between the levels of NK cell-mediated cytotoxicity from IL-2+anti-CD16 mAb-treated and IL-2-activated OC-expanded NK cells (Figure 4C and Appendix A). Similarly, we observed higher levels of NK cell-mediated ADCC by primary NK cells in comparison to OC-expanded NK cells when PL12 tumors were used as targets (Appendix A).

### 2.5. Higher Levels of NK Cell-Mediated Direct Cytotoxicity and ADCC by Freshly Isolated NK Cells When Compared to Either Parental NK92 Tumors and Those Expressing CD16 Receptor

Next, we explored the CD16 surface expression and the functional differences between primary human peripheral blood-derived NK cells, parental NK92, and the CD16-expressing NK92 tumors (NK92-176V). Primary NK cells exhibited a high surface expression of CD16 receptor as compared to NK92 or NK92-176V (Appendix A). We have also found lower levels of NK cell-mediated cytotoxicity and ADCC mediated by the NK92 parental and NK92-176V tumors than by primary NK cells when tested against untreated or anti-MICA/B mAb-treated PL12 tumors (Figure 5A). The levels of NK92 and NK92-176V-mediated direct cytotoxicity were lower than those of primary NK cells against MP2 tumors (Figure 5B). NK92-176V cells mediated higher levels of cytotoxicity against both PL12 and MP2 cells when compared to NK92 cells, although the effects of anti-MICA/B mAbs were not pronounced in either group (Figure 5).

### 2.6. Differentiated Tumor Cells Expressed Higher Levels of Surface EGFR and PDL-1 and Were More Susceptible to NK Cell-Mediated ADCC in the Presence of Anti-EGFR and Anti-PDL1 mAbs Compared to Their Stem-Like/Poorly Differentiated Counterparts

In addition to increase in MICA/B ligands on OSCCs, PL12, and NK-differentiated OSCSCs, we also observed higher levels of epidermal growth factor receptor (EGFR) (Figure 6A) and PDL1 receptors (Appendix A) on these tumors when compared to OSCSCs and MP2 tumors. Indeed, unlike OSCSCs, signaling through EGFR on OSCCs was able to increase the expression of phospho-STAT3 (signal transducer and activator of transcription 3) (Appendix A). IL-2-treated NK cells mediated significant ADCC against OSCCs in the presence of anti-EGFR mAbs, whereas these antibodies inhibited IL-2-treated NK cell-mediated ADCC against OSCSCs tumors (Figure 6B). Similarly, NK cells mediated significant ADCC against OSCCs, NK-differentiated OSCSCs, and PL12 in the presence of anti-PDL1 mAbs, whereas these antibodies inhibited NK cell-mediated ADCC against OSCSCs and MP2 tumors (Appendix A). Taken together, the data indicated that well-differentiated tumor cells expressed MICA/B, EGFR, and PDL1 and that the treatment of these tumors with their respective antibodies mediated ADCC by the NK cells.

## 3. Discussion

NK cells mediate their cytotoxic function against tumors through direct cytotoxicity and ADCC. A great number of antibodies made against distinct receptors on a variety of tumor cells are currently in clinical use. These antibodies not only inhibit various tumor cell functions by targeting specific receptors, they also guide NK cells to the targeted tumor cells to affect ADCC. Unfortunately, cancer patients who have dysfunctional NK cells also lack ADCC, as shown in this study, and the defect is in great part due to the decrease in CD16 expression in cancer patients. In a number of previous studies, it was shown that Adam 17 is an important enzyme that regulates the shedding of the CD16 receptor, and it can inhibit shedding, thereby increasing NK cell function [67,68]. Whether Adam 17 or any other effective enzyme could increase or restore NK cell-mediated ADCC in cancer patients awaits future investigations.

We have identified and characterized several oral and pancreatic tumor cell lines in different stages of differentiation [18,63,69,70]. Our previous studies have established OSCSCs as oral and MP2 as pancreatic stem-like/poorly differentiated tumors and OSCCs and PL12 tumors as well-differentiated oral and pancreatic tumors, respectively using CD44, CD54, MHC-class I, and PD-L1 surface antigens [62,70]. In this study, we established that the levels of MICA/B expressions were higher on the surfaces of well-differentiated tumor cells than on stem-like/poorly differentiated tumor cells. In accordance, untreated and IL-2-treated primary NK cells mediated significantly higher ADCC against anti-MICA/B mAb-treated OSCCs and PL12 tumors, and the addition of anti-CD16 mAbs to IL-2-activated NK cells abolished the increase in anti-MICA/B mAb-mediated ADCC (Figure 2C,E–G,K). On the other hand, no significant NK cell-mediated ADCC could be seen against OSCSCs or MP2 tumors that express no or lower levels of MICA/B, albeit NK cells mediated significant direct cytotoxicity against these tumors (Figure 2D,L). When OSCSCs or MP2 tumors were differentiated by NK supernatants and the levels of direct cytotoxicity and ADCC were measured, a significant increase in ADCC in the presence of negligible direct cytotoxicity (Figure 1D,G) was observed, which correlated with the increase in the surface expressions of MICA/B on the NK cell-differentiated OSCSCs and MP2 tumors. No increase in the levels of NK cell-mediated ADCC could be observed using patients’ NK cells, indicating a severe inhibition of ADCC. This is surprising, since even though there is a substantial decrease in CD16 expression on the surface of patients’ NK cells, there still exists a portion of the NK cells with decent levels of CD16 receptors. Whether there is also a functional deficiency of CD16 receptors in regard to ADCC in addition to decreased expression of this receptor on patients’ NK cells will require further studies.

The fold increase in ADCC was higher by untreated NK cells than those treated with IL-2. However, IFN-γ secretion was only induced in IL-2-treated NK cells during ADCC and not by the untreated NK cells, even though they mediated higher levels of ADCC. These results indicated the differential regulation of ADCC and IFN-γ secretion. In addition, untreated NK cells mediated no or slight direct killing in the majority of tumors tested. When NK cells were activated with IL-2, the levels of direct cytotoxicity increased, but the fold increase in ADCC was lower against OSCCs and PL12 tumors than those displayed by untreated NK cells (Figure 2I,N). These results indicated that once IL-2 triggers direct cytotoxicity, the levels of ADCC decreases, potentially indicating competition for CD16 receptors by the tumor cell ligands for direct killing as well as ADCC-mediated killing.

Unlike IL-2-activated primary NK cells, OC-expanded NK cells were found to mediate direct cytotoxicity against differentiated OSCCs and PL12 tumors (Figure 4A and Appendix A). When these tumors were used as targets of OC-expanded NK cells, no significant increase in ADCC could be observed; however, there were significant levels of direct cytotoxicity. Although there was a down-modulation of CD16 receptor on OC-expanded NK cells, the remaining amounts of CD16 were presumably sufficient to mediate ADCC. This observation further reinforces the possibility that putative tumor ligands or Fc regions of antibodies are likely engaged in a competitive manner in CD16 receptor binding, thereby decreasing the levels of ADCC while increasing cytotoxicity or vice versa. Indeed, OC-expanded NK cells have increased levels of NKG2D, which could directly bind to MICA/B and mediate cytotoxicity [61].

Primary NK cells exhibited a greater potential for ADCC than did the CD16-expressing NK92 cells, although CD16-expressing NK92 cells could also mediate ADCC to a much lower extent against MICA/B ligands. These differences can be due to the density of CD16 expression on primary NK cells as well as the superb ability of these cells to mediate direct cytotoxicity as well as ADCC. Indeed, NK92 cells do not mediate significant direct killing against OSCSCs and MP2 tumors ([71], and Figure 5B).

Similar to MICA/B, the levels of EGFR and PDL1 are increased on NK-differentiated OSCSCs and well-differentiated OSCCs and PL-12 tumors but not on OSCSCs or MP2 CSCs/poorly differentiated tumors (Figure 6A and Appendix A). Accordingly, the levels of NK cell-mediated ADCC using these antibodies were increased in NK-differentiated OSCSCs and in well differentiated OSCCs and PL-12 tumors. In contrast, there was a slight change or a decrease in direct cytotoxicity when antibodies were used in the cultures of NK cells with OSCSC and MP2 oral and pancreatic tumors. These results are in agreement with the findings obtained using anti-MICA/B mAbs. It is possible that a lack of expression of many key receptors on OSCSCs and MP2 tumors, as seen in this study, is a potential underlying mechanism for their aggressiveness. In addition, the lack of such receptor expression is likely to shield these tumors from receiving signals, which could potentially control their growth and expansion.

Overall, there is a possibility that CD16 receptors can be used in both direct cytotoxicity and in ADCC, resulting in competition for the use of the receptors depending on the differentiation status of the tumor cells and the stage of maturation of the NK cells. In fact, ligands other than the Fc portion of antibodies have previously been identified that can bind to the CD16 receptor [72,73,74]. Delineation of the ligands used for binding to CD16 and induction of direct cytotoxicity versus those used for ADCC should provide the basis for novel treatment strategies. Indeed, the addition of anti-CD16 receptor antibody also inhibits direct cytotoxicity as well as inhibition of ADCC.

## 4. Materials and Methods

### 4.1. Cell Lines, Reagents, and Antibodies

Oral squamous carcinoma cells (OSCCs) and oral squamous carcinoma stem cells (OSCSCs) were isolated from patients with tongue tumors at UCLA [64,75]. OSCCs, OSCSCs, and K562 tumors were cultured in RPMI 1640 (Life Technologies, Los Angeles, CA, USA) supplemented with 10% fetal bovine serum (FBS) (Gemini Bio-Product, West Sacramento, CA, USA). Mia PaCa-2 (MP2) and PL12 human pancreatic cell lines were provided by Drs. Guido Eibl and Nicholas Cacalano (UCLA David Geffen School of Medicine, Los Angeles, CA, USA). MP2 and PL12 cells were cultured in dulbecco’s modified eagle medium (DMEM) supplemented with 10% FBS and 2% penicillin–streptomycin (Gemini Bio-Products, West Sacramento, CA, USA). RPMI 1640 supplemented with 10% FBS was used to culture human NK cells. Alpha-minimum essential medium (α-MEM) (Life Technologies, Los Angeles, CA, USA) supplemented with 10% FBS was used for osteoclasts (OCs) cultures. Macrophage colony-stimulating factor (M-CSF), anti-CD16 mAb, and flow cytometric antibodies were purchased from Biolegend (San Diego, CA, USA). Receptor activator of nuclear factor kappa-B ligand (RANKL) was purchased from PeproTech (Cranbury, NJ, USA), and recombinant human IL-2 was obtained from NIH-BRB. Anti-MICA/B mAbs used for ADCC were a generous gift from Dr. Jennifer Wu (Feinberg School of Medicine, Northwestern University, Evanston, IL, USA). NK92 was obtained from ATCC (Baltimore, MD, USA) and maintained in Alpha-MEM medium without ribonucleosides and deoxyribonucleosides supplemented with 2 mM L-glutamine, 1.5 g/L sodium bicarbonate, 0.2 mM inositol, 0.1 mM 2-mercaptoethanol, 0.02 mM folic acid, 100–200 U/mL rh-IL-2, 10% horse serum, and 10% FBS. NK92-176V (CD16^high^ transfected NK92) was a generous gift from Dr. Kerry Campbell (FOX Chase Cancer Center). AJ2 is a combination of eight different strains of Gram-positive probiotic bacteria (*Streptococcus thermophiles*, *Bifidobacterium longum*, *Bifidobacterium breve*, *Bifidobacterium infantis*, *Lactobacillus acidophilus*, *Lactobacillus plantarum*, *Lactobacillus casei*, and *Lactobacillus bulgaricus*) selected for its superior ability to optimally induce the secretion of both pro-inflammatory and anti-inflammatory cytokines from NK cells [18]. RPMI 1640 supplemented with 10% FBS was used to re-suspend AJ2. Human ELISA kits for IFN-γ were purchased from Biolegend (San Diego, CA, USA). Phosphate-buffered saline (PBS) and bovine serum albumin (BSA) were purchased from Life Technologies (Los Angeles, CA, USA).

### 4.2. Purification of Human NK Cells and Monocytes

Written informed consents approved by the UCLA Institutional Review Board (IRB) were obtained from healthy donors and cancer patients, and all procedures were approved by the UCLA-IRB. Peripheral blood mononuclear cells (PBMCs) were isolated from peripheral blood as previously described [76]. Briefly, PBMCs were obtained after Ficoll-hypaque centrifugation and were used to isolate NK cells and monocytes using the EasySep^®^ Human NK cell and EasySep^®^ Human Monocytes enrichment kits, respectively, purchased from Stem Cell Technologies (Vancouver, BC, Canada). Isolated NK cells and monocytes were stained with anti-CD16 and anti-CD14 antibodies, respectively, to measure the cell purity using flow cytometric analysis.

### 4.3. NK Cells Induced Differentiation of OSCSCs and MP2 Tumors

Human NK cells were purified from healthy individuals’ PBMCs as described above. NK cells were treated with a combination of IL-2 (1000 U/mL) and anti-CD16mAbs (3 μg/mL) for 18 h, after which the supernatant was harvested and the levels of IFN-γ were assessed using single ELISA and later used in differentiation experiments. The differentiation of OSCSCs and MP2 cells was conducted with an average total of 2000 to 3500 pg and 5000 to 7000 pg of IFN-γ from IFN-γ containing supernatants, respectively, over a 7-day period, as previously described [64]. Initially, 1 × 10^6^ tumor cells were cultured and treated with NK supernatant for differentiation as described, after which tumor cells were rinsed with 1× PBS, detached, and used for experiments.

### 4.4. Generation of Human OCs

To generate OCs, monocytes were cultured in alpha-MEM media supplemented with M-CSF (25 ng/mL) for 21 days and RANKL (25 ng/mL) from day 6 to 21 days. The media were replenished every three days.

### 4.5. Sonication of Probiotic Bacteria (AJ2)

AJ2 bacteria were weighed and re-suspended in RPMI 1640 medium containing 10% FBS at a concentration of 10 mg/mL. The bacteria were thoroughly vortexed and sonicated on ice for 15 s at six to eight amplitudes. Then, sonicated samples were incubated for 30 s on ice, and the cycle was repeated for five rounds. After every five rounds of sonication, the samples were examined under the microscope until at least 80% of bacterial walls were lysed. It was determined that approximately 20 rounds of sonication/incubation on ice were necessary to achieve complete sonication. Finally, the sonicated AJ2 (sAJ2) were aliquoted and stored at −80 °C until use.

### 4.6. Expansion of NK Cells

Purified human NK cells were activated with rh-IL-2 (1000 U/mL) and anti-CD16 mAbs (3 µg/mL) for 18–20 h before they were co-cultured with osteoclasts (OCs) and sAJ2 (OCs:NK:sAJ2; 1:2:4) in RPMI 1640 medium containing 10% FBS. The media were refreshed every three days with RPMI complete medium containing rh-IL-2 (1500 U/mL).

### 4.7. Enzyme-Linked Immunosorbent Assays (ELISAs)

Single ELISAs were performed as previously described [76]. To analyze and obtain the cytokine and chemokine concentration, a standard curve was generated by either two- or three-fold dilutions of recombinant cytokines provided by the manufacturer.

### 4.8. ^51^Cr release Cytotoxicity Assay

The ^51^Cr release cytotoxicity assay was performed as previously described [77]. Briefly, different numbers of effector cells were incubated with ^51^Cr–labeled target cells. After a 4-h incubation period, the supernatants were harvested from each sample, and the released radioactivity was counted using a gamma counter. The percentage specific cytotoxicity was calculated as follows:(1)$$\%\mathrm{cytotoxicity}=\frac{\mathrm{Experimental}\;\mathrm{cpm}-\mathrm{spontaneous}\;\mathrm{cpm}}{\mathrm{Total}\;\mathrm{cpm}-\mathrm{spontaneous}\;\mathrm{cpm}}$$
where LU 30/10^6^ is calculated by using the inverse of the number of effector cells needed to lyse 30% of tumor target cells × 100.

### 4.9. Antibody-Dependent Cell-Mediated Cytotoxicity (ADCC) Measurements

Tumor cells (target cells) were ^51^Cr-labeled and were incubated for an hour, after which unbound ^51^Cr was washed. Then, cells (1 × 10^6^ cells/mL) were treated with anti-MICA/B mAbs (5 μg/mL) or anti-EGFR mAbs (10 µg/mL) or anti-PDL1 mAbs (20 µg/mL) for 30 min and washed with medium to remove excess unbound antibodies. Then, antibody-treated ^51^Cr-labeled cells were cultured with effector cells at various effector to target ratios, and the cytotoxicity against tumor cells was assessed using the ^51^Cr release cytotoxicity assay as described above.

### 4.10. Surface Staining Assay

For surface staining, the cells were washed twice using 1%BSA/PBS. Predetermined optimal concentrations of specific human monoclonal antibodies were added to 1 × 10^4^ cells in 50 µL of 1%BSA/PBS and were incubated on ice for 30 min. Thereafter, cells were washed in 1%BSA/PBS and brought to 500 µL with 1%BSA/PBS. Flow cytometric analysis was performed using the Beckman Coulter Epics XL cytometer (Brea, CA, USA), and the results were analyzed in the FlowJo vX software (Ashland, OR, USA).

### 4.11. Western Blot

OSCCs and OSCSCs tumor cells were lysed in a lysis buffer containing 50 mM Tris-HCL (pH 7.4), 150 mM NaCl, 1% Nonidet P-40 (*v/v*), 1 mM sodium orthovanadate, 0.5 mM ethylenediaminetetraactetic acid (EDTA), 10 mM NaF, 2 mM phenylmethylsulfonyl fluoride (PMSF), 10 µg/mL leupeptin, and 2 U/mL aprotinin for 15 min on ice. Then, the samples were sonicated for 3 s. The tumor cell lysates were centrifuged at 14,000 rpm for 10 min, and the supernatants were removed and the levels of protein were quantified by the Bradford method. The cell lysates were denatured by boiling in 5× SDS sample buffer. Equal amounts of cell lysates were loaded onto 10% SDS-PAGE and transferred onto Immobilon-P membranes (Millipore, Billerica, MA, USA). The membranes were blocked with 5% non-fat milk in PBS plus 0.1% Tween-20 for 1 h. Primary antibodies at the predetermined dilutions were added for 1 h at room temperature. Then, membranes were incubated with 1:1000 dilution of horseradish peroxidase-conjugated secondary antibody. Blots were developed by enhanced chemiluminescence (ECL purchased from Pierce Biotechnology, Rockford, IL, USA).

### 4.12. Statistical Analyses

All statistical analyses were performed using the GraphPad Prism-8 software. An unpaired or paired, two-tailed Student’s *t*-test was performed for experiments with two groups. One-way ANOVA with a Bonferroni post-test was used to compare different groups for experiments with more than two groups. (*n*) denotes the number of human donors or the number of samples for each experimental condition. Duplicate or triplicate samples were used in the in vitro studies for assessment. The following symbols represent the levels of statistical significance within each analysis: **** (*p* value < 0.0001), *** (*p* value < 0.001), ** (*p* value 0.001–0.01), * (*p* value 0.01–0.05)

## 5. Conclusions

In this study we provided evidence that NK cells mediate lysis of both stem-like/poorly differentiated tumors and well-differentiated tumors via direct cytotoxicity and ADCC, respectively. By using antibodies to MICA/B, EGFR and PDL1 surface receptors expressed on well-differentiated but not on stem-like/poorly differentiated tumors we demonstrated significant NK cell mediated ADCC in the absence of direct killing. In addition, our results suggested the possibility of CD16 receptors mediating both direct cytotoxicity and ADCC, resulting in the competitive use of these receptors in either direct killing or antibody dependent cellular cytotoxicity, depending on the differentiation status of tumor cells and the stage of maturation and activation of NK cells. Thus, these two modes of NK cell mediated killing ensures that both CSC/poorly differentiated and well-differentiated tumors are eliminated.

## Figures and Tables

**Figure 1 cancers-13-00239-f001:**
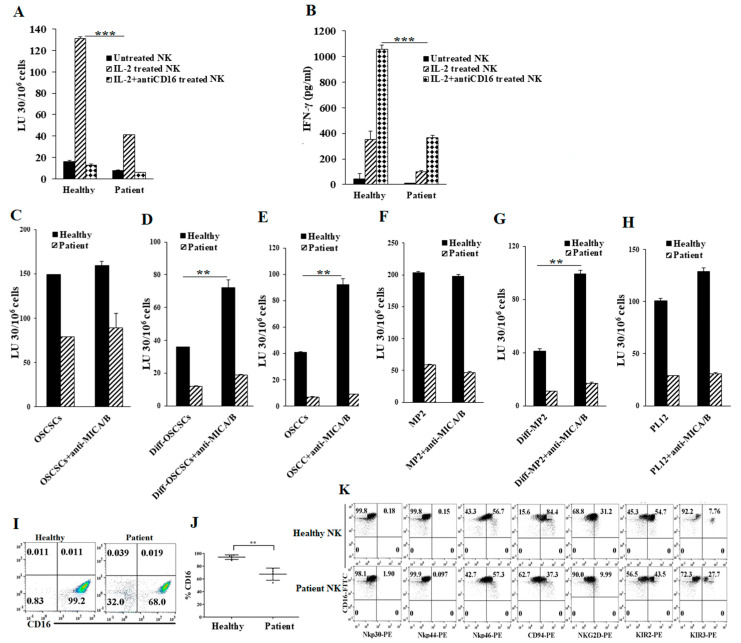
Cancer patients’ natural killer (NK) cells exhibit lower functional activity and NK cell-mediated antibody-dependent cell cytotoxicity (ADCC) when compared to healthy individuals’ NK cells. Purified NK cells (1 × 10^6^ cells/mL) from healthy individuals and pancreatic cancer patients were left untreated, treated with interleukin-2 (IL-2) (1000 U/mL), or treated with a combination of IL-2 (1000 U/mL) and anti-CD16 mAb (3 µg/mL) for 18 h and were added to ^51^Cr-labeled oral stem-like/poorly-differentiated oral squamous cancer stem cells (OSCSCs) at various effector-to-target ratios. NK cell-mediated cytotoxicity was measured using a standard 4-h ^51^Cr release assay against OSCSCs. The lytic units (LU) 30/10^6^ cells were determined using the inverse number of NK cells required to lyse 30% of target cells × 100 (**A**) *** (*p* value < 0.001). NK cells were isolated and prepared as described in Figure 1A for 18 h before the supernatants were harvested and the levels of IFN-γ secretion were determined using single ELISA (**B**) *** (*p* value < 0.001). One of 20 experiments is shown in Figure 1A,B. Purified NK cells (1 × 10^6^ cells/mL) from healthy individuals and cancer patients were treated with IL-2 (1000 U/mL) for 18 h and were used as effectors in ^51^Cr release assay. OSCSCs and Mia PaCa-2 (MP2) tumors were differentiated as described in the Materials and Methods. OSCSCs (**C**), NK cell-differentiated OSCSCs (**D**)**,** oral squamous carcinoma cells (OSCCs) (**E**), MP2 (**F**), NK cell-differentiated MP2 (**G**)**,** and PL12 cells (**H**) were labeled with ^51^Cr for an hour, after which cells were washed to remove unbound ^51^Cr. Then, ^51^Cr-labeled tumor cells were left untreated or treated with anti-major histocompatibility complex-class I chain related proteins A and B (MICA/B) monoclonal antibodies (mAbs) (5 μg/mL) for 30 min. The unbounded antibodies were washed away, and the cytotoxicity against the tumor cells was determined using a standard 4-h ^51^Cr release assay. LU 30/10^6^ cells were determined as described in Materials and Methods (**C–H**) ** (*p* value 0.001–0.01). The surface expression levels of CD16 of freshly purified NK cells from healthy individuals and cancer patients were analyzed using flow cytometry. IgG2 isotype antibodies were used as controls (*n* = 4) (**I,J**) ** (*p* value 0.001–0.01). Freshly purified NK cells from healthy individuals and cancer patients were analyzed for the surface expression levels of CD16, Nkp30, Nkp44, Nkp46, CD94, NKG2D, KIR2, and KIR3 using flow cytometry. IgG2 isotype antibodies were used as controls (**K**). One of eight representative experiments is shown in Figure 1K.

**Figure 2 cancers-13-00239-f002:**
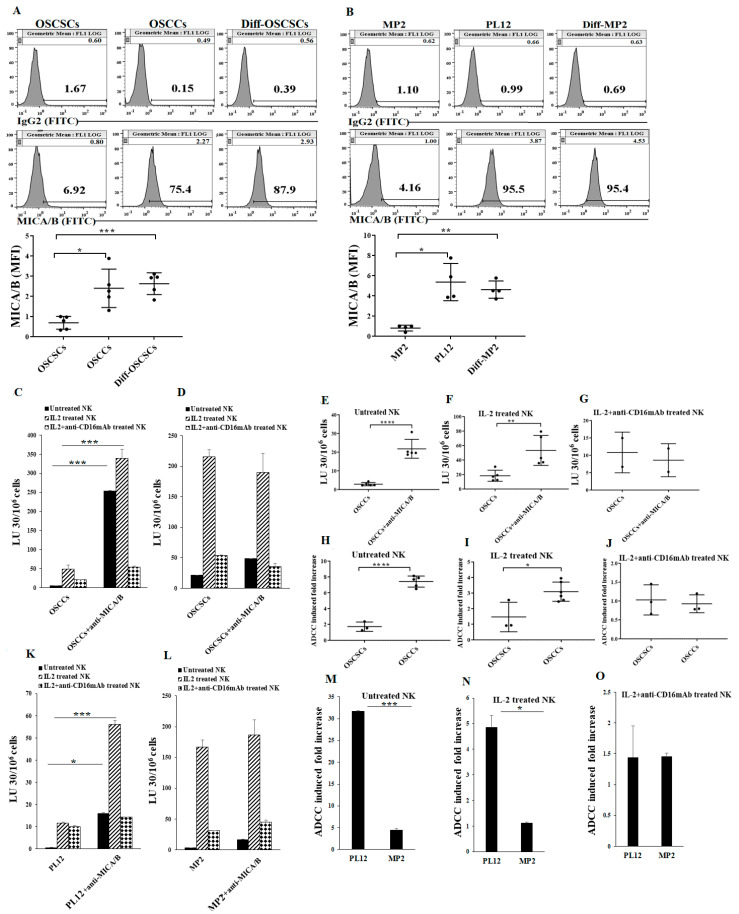
Differentiated tumors expressed higher surface levels of MICA/B and were more susceptible to NK cell-mediated cytotoxicity when compared to their stem-like counterparts in the presence of anti-MICA/B mAbs. OSCSCs were differentiated as described in the Materials and Methods. The surface expression levels of MICA/B on OSCSCs, OSCCs, and NK cell-differentiated OSCSCs were assessed using flow cytometric analysis. IgG2 isotype antibodies were used as controls (*n* = 5) (**A**). MP2 were differentiated as described in the Materials and Methods. The surface expression levels of MICA/B on MP2, PL12, and NK cell-differentiated MP2 were assessed using flow cytometric analysis. IgG2 isotype antibodies were used as controls (*n* = 4) (**B**). Freshly purified NK cells were left untreated, treated with IL-2 (1000 U/mL), or treated with a combination of IL-2 and anti-CD16 mAb (3 μg/mL) for 18 h and were used as effectors against OSCCs (**C**) and OSCSCs (**D**) to measure NK cell-mediated ADCC as described in Materials and Methods (*n* = 6) (**C,D**). The NK cell-mediated ADCC was measured using untreated (*n* = 5) (**E**), IL-2 treated (*n* = 5) (**F**), and IL-2 + anti-CD16 mAb treated (*n* = 2) (**G**) NK cells as effectors against target OSCCs (**E–G**). Fold increase in ADCC against OSCCs and OSCSCs mediated by untreated (**H**) (*n* = 5), IL-2 treated (**I**) (*n* = 5), and IL-2+anit-CD16 mAbs (**J**) (*n* = 3) were calculated. NK cells were prepared as described in Figure 2C and were used as effectors to measure NK cell-mediated ADCC against PL12 (**K**) and MP2 tumors (**L**). Fold increases in NK cell-mediated ADCC against MP2 and PL12 tumors by untreated (**M**), IL-2 treated (**N**), and IL-2+anti-CD16 mAb treated (**O**) NK cells were calculated. **** (*p* value < 0.0001), *** (*p* value < 0.001), ** (*p* value 0.001–0.01), * (*p* value 0.01–0.05)

**Figure 3 cancers-13-00239-f003:**
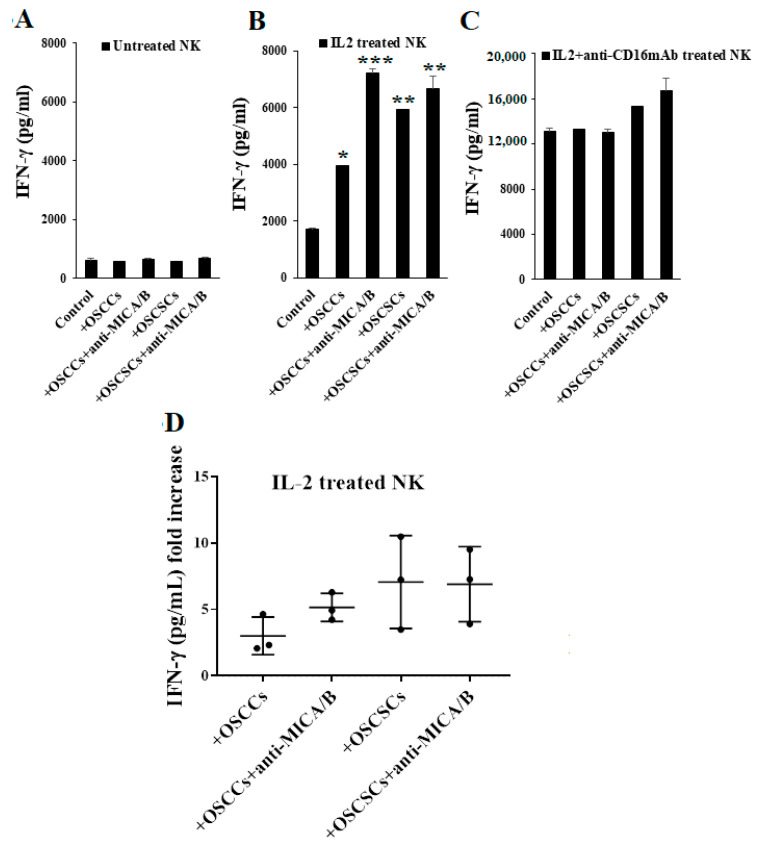
Differentiated tumor cells treated with anti-MICA/B mAb increased IFN-γ secretion of IL-2 activated NK cells. Freshly purified NK cells from healthy individuals were left untreated (**A**), treated with IL-2 (1000 U/mL) (**B**), or treated with a combination of IL-2 and anti-CD16 mAb (3 μg/mL) (**C**) for 18 h. OSCCs and OSCSCs were treated with anti-MICA/B mAb (5 μg/mL) for 18 h, excess unbounded antibodies were removed, and the tumor cells were co-cultured with NK cells. The supernatants were harvested from the co-cultures after 24 h, and the concentrations of IFN-γ were determined using single ELISA (*n* = 2) (**A–C**). The ratios of IFN-γ secretion of IL-2 activated NK cells induced by untreated or anti-MICA/B mAb treated oral tumor cells (OSCCs and OSCSCs) were determined as fold increase (*n* = 3) (**D**). *** (*p* value < 0.001), ** (*p* value 0.001–0.01), * (*p* value 0.01–0.05).

**Figure 4 cancers-13-00239-f004:**
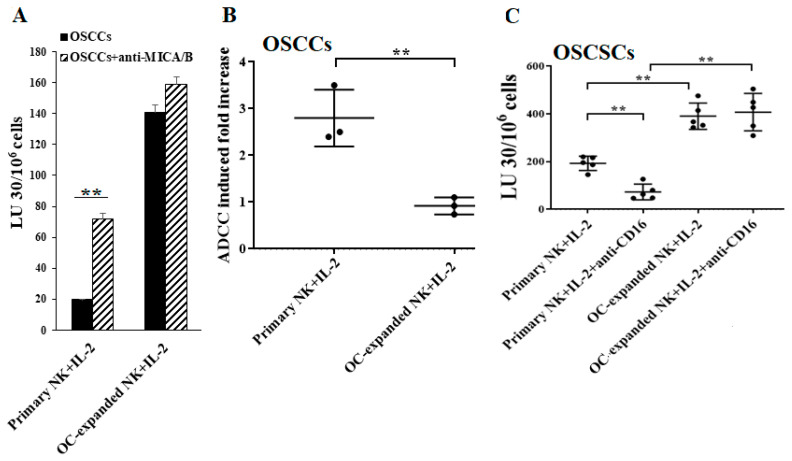
Higher levels of NK cell-mediated ADCC were seen in freshly isolated primary NK cells vs. in osteoclast (OC)-expanded NK cells. OCs were generated and OC-expanded NK cells were prepared as described in the Materials and Methods. Freshly purified primary and OC-expanded NK cells were both treated with IL-2 (1000 U/mL) for 18 h and were used as effector cells to measure NK cell-mediated ADCC against OSCCs as described in Materials and Methods (**A**). Fold increases in ADCC were calculated (**B**). Freshly purified primary and OC-expanded NK cells were treated with IL-2 (1000 U/mL) or a combination of IL-2 (1000 U/mL) and anti-CD16 mAbs (3 µg/mL) for 18 h before being used as effector cells to measure NK cell-mediated ADCC against OSCSCs as described in the Materials and Methods (*n* = 5) (**C**). ** (*p* value 0.001–0.01).

**Figure 5 cancers-13-00239-f005:**
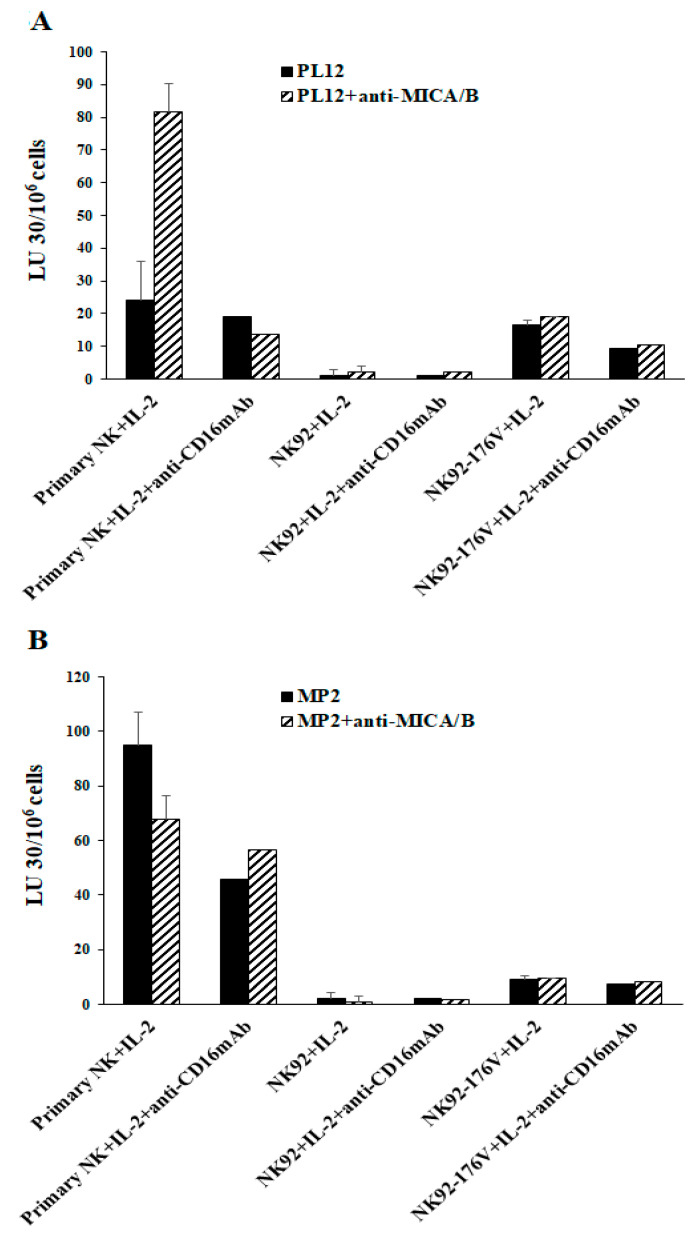
Higher levels of NK cell-mediated ADCC were seen in freshly isolated primary NK cells vs. in NK92 and NK92-176V. Freshly purified primary NK cells, NK92, and NK92-176V cells were either treated with IL-2 (1000 U/mL) or a combination of IL-2 and anti-CD16 mAbs (3 μg/mL) for 18 h and were used as effector cells to measure NK cell-mediated ADCC against PL12 (**A**) and MP2 tumors (**B**), as described in Materials and Methods.

**Figure 6 cancers-13-00239-f006:**
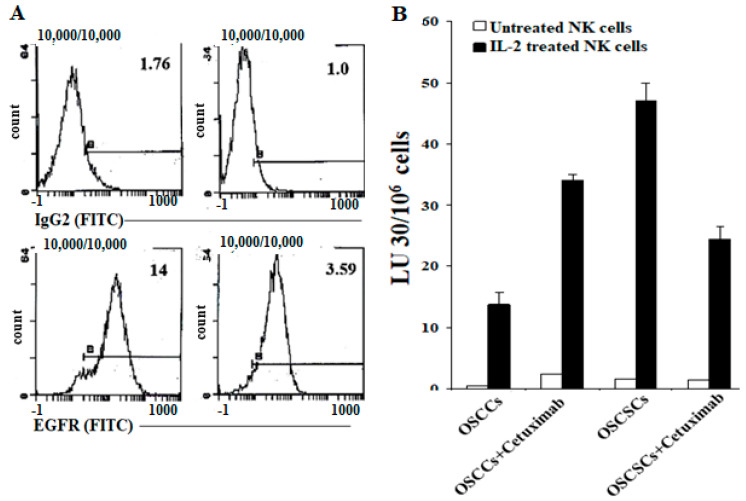
Higher levels of epidermal growth factor receptor (EGFR) surface expressions and NK cell-mediated ADCC in OSCCs in comparison to OSCSCs. The levels of EGFR expression were determined on OSCCs and OSCSCs using surface staining with Cetuximab (**A**). Numbers in the histograms represent mean fluorescence intensity (MFI). Purified NK cells (1 × 10^6^ cells/mL) were left untreated or treated with IL-2 (1000 U/mL) for 18 h and used against untreated and Cetuximab-treated OSCCs and OSCSCs in a 4-h ^51^Cr release assay. The lytic units (LU) 30/10^6^ cells were determined using the inverse number of NK cells required to lyse 30% of target cells × 100 (**B**).

## Data Availability

The data presented in this study are available in the article or Appendix A of “ADCC against MICA/B is Mediated against Differentiated Oral and Pancreatic and Not Stem-Like/Poorly Differentiated Tumors by the NK Cells; Loss in Cancer Patients due to Down-Modulation of CD16 Receptor”.

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
