# Peer review of "ADCC against MICA/B Is Mediated against Differentiated Oral and Pancreatic and Not Stem-Like/Poorly Differentiated Tumors by the NK Cells; Loss in Cancer Patients due to Down-Modulation of CD16 Receptor"

_cancers, 2021, doi:10.3390/cancers13020239_

Round 1

Reviewer 1 Report

This intersting paper from Kaur et al. describe different sensitivity of NK cells depending on the differentiation state of pancreatic and oral tumor cells. Here are my comments :

  • To evaluate Cancer patients' NK activity (Fig 1A/B), why the autors have combined antiCD16 + IL-2 to activate the cells? It would have been better to show only the untreated NK and the IL2 treated condition and discuss the fact that the differences are seen only for activated NK Cells. Otherwise, authors should explain why aCD16 treatment increase IFNg production but decrease cytotoxicity. It would also be interesting to add a control condition like K562 cells in panel 1A and 1B
  • Line 94-95. Autors said that they "observed very little or no NK cell mediated ADCC aigainst tumor cells". They should add that in healthy donors, ADCC is observed only in differentiate tumor cells (1D/1E/1G/1H) and not for stem like cells (1C/1H).
  • Autors should comment Fig 1I, 1J, 1K and rely the loss of ADCC with the dicrease of CD16 expression
  • Fig2A and 2B: please indicate the MFI of MICA/B and not the %of positive cells on the FACS plots
  • Fig 4 and 5 : Can autors compare the expression of CD16 and NKG2D in Primary NK/OC-expanded NK and NK92 (+IL2 or IL2/CD16) to help understand the data?

Author Response

We appreciate the kind comments from reviewer 1 stating that “This interesting paper from Kaur et al. describe different sensitivity of NK cells depending on the differentiation state of pancreatic and oral tumor cells”. Please find the point by point answers to the reviewer comments below:

  • To evaluate Cancer patients' NK activity (Fig 1A/B), why the authors have combined antiCD16 + IL-2 to activate the cells? It would have been better to show only the untreated NK and the IL2 treated condition and discuss the fact that the differences are seen only for activated NK Cells. Otherwise, authors should explain why aCD16  treatment increase IFNg production but decrease cytotoxicity. It would also be interesting to add a control condition like K562 cells in panel 1A and 1B

Response: We appreciate the comment, and the reason we added IL-2+anti-CD16 is due to the fact that ADCC by NK cells is known to be primarily mediated by the CD16 receptor binding to the Fc portion of the antibody bound to the tumor cells. Thus we wanted to show that blocking CD16 will inhibit ADCC providing the proof that ADCC is mediated by the action of CD16. This finding is not only provided by our studies but those of the others [1-4].

In addition, we have established the concept of split anergy in NK cells a long time ago and written many manuscripts and reviews describing the concept [5, 6]. In split anergy NK cells lose ability to kill but maintain the ability to secrete IFN-g which mediates differentiation of tumors [5, 7, 8].  

As suggested by the reviewer we have now added K562 results in the supplementary file, Fig. S1.

  • Line 94-95. Authors said that they "observed very little or no NK cell mediated ADCC aigainst tumor cells". They should add that in healthy donors, ADCC is observed only in differentiated tumor cells (1D/1E/1G/1H) and not for stem like cells (1C/1H).

Response: We added the suggested clarification. Thank you

  • Authors should comment Fig 1I, 1J, 1K and rely the loss of ADCC with the decrease of CD16 expression

Response: We added the suggested clarification. Thank you

  • Fig2A and 2B: please indicate the MFI of MICA/B and not the %of positive cells on the FACS plots

Response: We would like to point that the MFI were shown in top right corner on FACS plots and also in scatter plot graphs in the original manuscript.

  • Fig 4 and 5 : Can authors compare the expression of CD16 and NKG2D in Primary NK/OC-expanded NK and NK92 (+IL2 or IL2/CD16) to help understand the data?

Response: We have previously published the data on CD16 and NKG2D comparison of primary NK and OC-expanded NK cells [9](Kaur K et al. Front Immunol. 2017 Apr 5;8:297. doi: 10.3389/fimmu.2017.00297).

CD16 surface expression on NK92 and NK92-176V was previously published in [10]“Magister et al, 2015; Oncotarget, Vol. 6, No. 26”. For more details and extensive characterization of these cells please refer to the published manuscript.

However, to facilitate the review we are presenting these data also in the supplementary file (Fig. S5) now.

  1. Yeap, W.H., et al., CD16 is indispensable for antibody-dependent cellular cytotoxicity by human monocytes. Scientific Reports, 2016. 6(1): p. 34310.
  2. Bhatnagar, N., et al., FcgammaRIII (CD16)-mediated ADCC by NK cells is regulated by monocytes and FcgammaRII (CD32). Eur J Immunol, 2014. 44(11): p. 3368-79.
  3. Oboshi, W., et al., The influence of NK cell-mediated ADCC: Structure and expression of the CD16 molecule differ among FcgammaRIIIa-V158F genotypes in healthy Japanese subjects. Hum Immunol, 2016. 77(2): p. 165-71.
  4. Nimmerjahn, F. and J.V. Ravetch, Fcgamma receptors as regulators of immune responses. Nat Rev Immunol, 2008. 8(1): p. 34-47.
  5. Tseng, H.C., N. Cacalano, and A. Jewett, Split anergized Natural Killer cells halt inflammation by inducing stem cell differentiation, resistance to NK cell cytotoxicity and prevention of cytokine and chemokine secretion. Oncotarget, 2015. 6(11): p. 8947-59.
  6. Jewett, A., et al., Rapid and potent induction of cell death and loss of NK cell cytotoxicity against oral tumors by F(ab')2 fragment of anti-CD16 antibody. Cancer Immunol Immunother, 2008. 57(7): p. 1053-66.
  7. Bonavida, B., L.T. Lebow, and A. Jewett, Natural killer cell subsets: maturation, differentiation and regulation. Nat Immun, 1993. 12(4-5): p. 194-208.
  8. Kaur, K., et al., Natural killer cells target and differentiate cancer stem-like cells/undifferentiated tumors: strategies to optimize their growth and expansion for effective cancer immunotherapy. Curr Opin Immunol, 2018. 51: p. 170-180.
  9. Kaur, K., et al., Novel Strategy to Expand Super-Charged NK Cells with Significant Potential to Lyse and Differentiate Cancer Stem Cells: Differences in NK Expansion and Function between Healthy and Cancer Patients. Front Immunol, 2017. 8: p. 297.
  10. Magister, Š., et al., Regulation of split anergy in natural killer cells by inhibition of cathepsins C and H and cystatin F. Oncotarget, 2015. 6(26): p. 22310-27.

Reviewer 2 Report

This manuscript claims that tumor cells (e.g. oral squamous and pancreatic cancer) differentially express the NKG2D ligands MICA/B depending on their differentiation stage and thereby affect their susceptibility to NK cells in the context of natural cytotoxicity vs antibody-dependent cellular cytotoxicity (ADCC). They compared the ability of different NK cells (freshly isolated, IL-2 activated, or osteoclast-expanded NK cells) to lyse tumor cells in direct cytotoxicity or anti-MICA/B mAb-mediated killing as a model of ADCC. They found that tumor cells that are differentiated or rendered to be differentiated by supernatant from NK cells upregulated MICA/B and are preferably killed by ADCC rather than natural cytotoxicity in the presence of anti-MICA/B mAb. In addition, this phenomenon was not observed when using osteoclast-expanded NK cells as effector cells that show superior natural cytotoxicity. Based on this finding, they conclude that natural cytotoxicity and ADCC against tumor cells are used in competitive manner depending on the differentiation status of tumor cells and activation potential of NK cells. Overall, the concept of this topic is of interest, their claims are not sufficiently supported by their results provided. Moreover, the experimental design appears irrelevant by targeting NKG2D ligand MICA/B rather than tumor antigen as to compare between natural cytotoxicity and ADCC.

Specific comments

MICA/B is the well-known ligands for NKG2D mediating robust natural cytotoxicity. Thus, it is of course that natural cytotoxicity and ADCC were used by NK cells in competitive manner as they used mAb targeting MICA/B. In this situation, it would be desirable to target tumor antigen such as anti-EGFR mAb to trigger ADCC and then compare the capacity of NK cells to lyse tumor cells by natural cytotoxicity compared with ADCC.

It is already known that certain ligands for NKG2D including MICA/B are downregulated or absent on undifferentiated or cancer stem cells as a mechanism of immune evasion (Paczulla AM et al., Nature, 2019, 572:254). Thus, the authors would like to assess the levels of ULBPs, another NKG2D ligands, or the ligands for other activating receptors to see if the upregulation of MICA/B upon tumor differentiation is confined to the regulation of MICA/B.

The references for Introduction is too much and needs to be selected to appropriate numbers. Moreover, certain references are irrelevant, inaccurate, redundant and missed, which needs to be corrected. For example, Refs. 18-20 are unrelated to MICA/B but to NCR and CD16. Ref. 40 is an Abstract but not a paper. Ref. 47 is missing. The same Refs are repeatedly used (11 and 27; 25 and 26; 55 and 83; 81 and 89 and 96; 84 and 94).

Define the type of cancer for NK cells in the figure 1 and/or its legend.

Figures 1K and 2M-O are the representative result. I am wondering if not the single experiment, then the results are to be presented with the statistical graph.

The manuscript contains some grammatical errors and sometimes is not easy to follow. Thus it is better to have Editing service to correct some mistakes and errors and make the manuscript more concise.

Author Response

This manuscript claims that tumor cells (e.g. oral squamous and pancreatic cancer) differentially express the NKG2D ligands MICA/B depending on their differentiation stage and thereby affect their susceptibility to NK cells in the context of natural cytotoxicity vs antibody-dependent cellular cytotoxicity (ADCC). They compared the ability of different NK cells (freshly isolated, IL-2 activated, or osteoclast-expanded NK cells) to lyse tumor cells in direct cytotoxicity or anti-MICA/B mAb-mediated killing as a model of ADCC. They found that tumor cells that are differentiated or rendered to be differentiated by supernatant from NK cells upregulated MICA/B and are preferably killed by ADCC rather than natural cytotoxicity in the presence of anti-MICA/B mAb. In addition, this phenomenon was not observed when using osteoclast-expanded NK cells as effector cells that show superior natural cytotoxicity. Based on this finding, they conclude that natural cytotoxicity and ADCC against tumor cells are used in competitive manner depending on the differentiation status of tumor cells and activation potential of NK cells.

Overall, the concept of this topic is of interest, their claims are not sufficiently supported by their results provided. Moreover, the experimental design appears irrelevant by targeting NKG2D ligand MICA/B rather than tumor antigen as to compare between natural cytotoxicity and ADCC.

Response: We hope that the rationale provided below will sufficiently explain the results. In addition, if we understand the reviewer correctly we have presented another EGF receptor targeting as target of NK ADCC as suggested by the reviewer. However, MICA/B also serves as an ADCC target as well as PDL-1 (not presented in this paper. If required we can present the findings too).

Specific comments

MICA/B is the well-known ligands for NKG2D mediating robust natural cytotoxicity. Thus, it is of course that natural cytotoxicity and ADCC were used by NK cells in competitive manner as they used mAb targeting MICA/B. In this situation, it would be desirable to target tumor antigen such as anti-EGFR mAb to trigger ADCC and then compare the capacity of NK cells to lyse tumor cells by natural cytotoxicity compared with ADCC.

Response: As stated above we are now showing the EGFR mediated ADCC by anti-EGFR-mAbs with both OSCC and OSCSC tumor cells as suggested by the reviewer. We are now presenting anti-EGFR results in the supplementary file in Fig. S4. As shown in the figure, ADCC is observed in differentiated OSCCs but not in OSCSCs since there is lower expression of EGFR on OSCSCs when compared to OSCCs and NK cells exhibit significant ADCC against OSCCs with higher expression of EGFR. On the other hand, NK cells mediate direct cytotoxicity against OSCSCs and addition of anti-EGFR receptor decreases the direct cytotoxicity. Similar results to those seen with anti-MICA/B mAbs.

It is already known that certain ligands for NKG2D including MICA/B are downregulated or absent on undifferentiated or cancer stem cells as a mechanism of immune evasion (Paczulla AM et al., Nature, 2019, 572:254).

Response: With due respect to our esteemed reviewer, we published on the finding of MICA/B expression on stem cells vs. differentiated cells in 2017 [1]. Therefore, our findings precede much earlier than the publication in Nature. We have indeed written many reviews on our views of how NK cells target and interact with stem cells and differentiated tumors, and the scape of cancer stem cells from elimination is one reason why these cells seed and metastasize in cancer patients since these patients don’t have effective mode of immune function as presented in this paper and in our previous publications in terms of direct killing [2-4].

Thus, the authors would like to assess the levels of ULBPs, another NKG2D ligands, or the ligands for other activating receptors to see if the upregulation of MICA/B upon tumor differentiation is confined to the regulation of MICA/B.

Response: We have determined the levels of ULBPs on OSCSCs and found similar to MICA/B they also expressed very low levels of ULBPs. To the best of our knowledge no laboratory has shown NK cell mediated ADCC against ULBP expressing targets. Because OSCCs express higher levels of EGFR as well as MICA/B, it is likely that they will also express more of ULBPs since there was a study demonstrating that tumor cell susceptibility to rituximab-induced ADCC was correlated by the amounts of NKG2D ligands expressed, such as ULBP 1–3 [5]. Our study also agrees with this study. However, the focus of our current paper is on MICA/B.

The references for Introduction is too much and needs to be selected to appropriate numbers. Moreover, certain references are irrelevant, inaccurate, redundant and missed, which needs to be corrected. For example, Refs. 18-20 are unrelated to MICA/B but to NCR and CD16. Ref. 40 is an Abstract but not a paper. Ref. 47 is missing. The same Refs are repeatedly used (11 and 27; 25 and 26; 55 and 83; 81 and 89 and 96; 84 and 94).

Response: We changed the introduction substantially and removed all duplicated/irrelevant references and corrected the others mentioned in the reviewer comments. Thank you

Define the type of cancer for NK cells in the figure 1 and/or its legend.

Response: We have defined the type of cancer in the figure legend.

Figures 1K and 2M-O are the representative result. I am wondering if not the single experiment, then the results are to be presented with the statistical graph.

Response: We are now presenting 1K with statistical analysis as Fig. S3, and also presented 2M-2O with statistical analysis.

The manuscript contains some grammatical errors and sometimes is not easy to follow. Thus it is better to have Editing service to correct some mistakes and errors and make the manuscript more concise.

Response: We have edited the manuscript substantially and hope that the revised version is satisfactory now.

  1. Kozlowska, A.K., et al., Differentiation by NK cells is a prerequisite for effective targeting of cancer stem cells/poorly differentiated tumors by chemopreventive and chemotherapeutic drugs. J Cancer, 2017. 8(4): p. 537-554.
  2. Kaur, K., et al., Novel Strategy to Expand Super-Charged NK Cells with Significant Potential to Lyse and Differentiate Cancer Stem Cells: Differences in NK Expansion and Function between Healthy and Cancer Patients. Front Immunol, 2017. 8: p. 297.
  3. Kaur, K., et al., Probiotic-Treated Super-Charged NK Cells Efficiently Clear Poorly Differentiated Pancreatic Tumors in Hu-BLT Mice. Cancers (Basel), 2019. 12(1).
  4. Kaur, K., et al., Super-charged NK cells inhibit growth and progression of stem-like/poorly differentiated oral tumors in vivo in humanized BLT mice; effect on tumor differentiation and response to chemotherapeutic drugs. Oncoimmunology, 2018. 7(5): p. e1426518.
  5. Inagaki, A., et al., Expression of the ULBP ligands for NKG2D by B-NHL cells plays an important role in determining their susceptibility to rituximab-induced ADCC. Int J Cancer, 2009. 125(1): p. 212-21.

Round 2

Reviewer 2 Report

The authors addressed my concerns raised to a reasonable extent.

Although I understand the situation, it would be better to show the results on anti-EGFR in more detail in the main manuscript rather than supplement, which would be more relevant to pathological setting.

In the revised manuscript, there is still duplication of reference (16 & 17), which needs to be corrected.

Author Response

The authors addressed my concerns raised to a reasonable extent.

Response: Thank you very much, we really appreciate your time and efforts.

Although I understand the situation, it would be better to show the results on anti-EGFR in more detail in the main manuscript rather than supplement, which would be more relevant to pathological setting.

Response: We appreciate the comment, and added anti-EGFR results in the main manuscript (Fig. 6). In addition, we added anti-PDL1 results in the supplementary file (Fig. S6)

In the revised manuscript, there is still duplication of reference (16 & 17), which needs to be corrected.

Response: We removed the duplicated references. Thank you